# Colistin Heteroresistance in *Klebsiella Pneumoniae* Isolates and Diverse Mutations of PmrAB and PhoPQ in Resistant Subpopulations

**DOI:** 10.3390/jcm8091444

**Published:** 2019-09-11

**Authors:** Hae Suk Cheong, So Yeon Kim, Yu Mi Wi, Kyong Ran Peck, Kwan Soo Ko

**Affiliations:** 1Division of Infectious Disease, Department of Internal Medicine, Kangbuk Samsung Hospital, Sungkyunkwan University School of Medicine, Seoul 03181, Korea; 2Department of Molecular Cell Biology, Sungkyunkwan University School of Medicine, Suwon 16419, Korea; 3Division of Infectious Diseases, Samsung Changwon Hospital, Sungkyunkwan University School of Medicine, Changwon 51353, Korea; 4Division of Infectious Disease, Samsung Medical Center, Sungkyunkwan University School of Medicine, Seoul 06351, Korea

**Keywords:** *Klebsiella pneumoniae*, colistin, heteroresistance, population analysis profiles, PmrAB, PhoPQ

## Abstract

Heteroresistance may pose a threat to the prognosis of patients following colistin treatment. We investigated colistin heteroresistance in *Klebsiella pneumoniae* isolates from South Korea. Among 252 *K. pneumoniae* blood isolates, 231 were susceptible to polymyxins. Heteroresistance to colistin was determined using population analysis profiles, disk diffusion assays, and E-test strip tests for the susceptible isolates. As a result, we identified three colistin-heteroresistant *K. pneumoniae* isolates belonging to separate clones (ST11, ST461, and ST3217) by multilocus sequence typing analysis. Two colistin-resistant subpopulations were selected from each heteroresistant isolate in either disk diffusion testing or E-testing. Two resistant subpopulations from the same isolate exhibited different amino acid substitutions in the two-component regulatory systems PmrAB and PhoPQ. An in vitro time–kill assay showed that meropenem combined with colistin had a 1× minimum inhibitory concentration bactericidal effect against a multidrug-resistant, colistin-heteroresistant isolate.

## 1. Introduction

*Klebsiella pneumoniae* is one of the most clinically significant pathogens belonging to the family *Enterobacteriaceae*. It is an important pathogen in community- and hospital-acquired infections [1]. Carbapenems have been used for infections caused by extended-spectrum β-lactamase (ESBL)-producing *K. pneumoniae*, and carbapenem-resistant *K. pneumoniae* first emerged in 1985 [2]. The prevalence of carbapenem-resistant *K. pneumoniae* has continued to increase globally, and such infections pose a critical threat to human health, with high mortality rates owing to the limited treatment options available [3,4,5]. Colistin and polymyxin B are important therapeutic options for treating infections caused by carbapenem-resistant *K. pneumoniae* [6,7].

Colistin exerts bactericidal action against Gram-negative pathogens, targeting the lipid A moiety of lipopolysaccharide (LPS) and leading to cell membrane disruption [8]. Unfortunately, colistin resistance has been reported in surveillance studies as well as in clinical case reports [9]. Resistance to colistin generally involves mutations in chromosomal genes. Acquired resistance to polymyxins in strains such as *K. pneumoniae* and *Escherichia coli* involves mutations in the two-component regulatory systems PmrAB and PhoPQ or alterations to the negative regulator of PhoPQ, MgrB [10,11].

Antibiotic heteroresistance is a phenomenon where subpopulations of seemingly isogenic bacteria exhibit a range of susceptibilities to a particular antibiotic [12]. It has only been reported in a limited number of studies because it cannot be assessed using ordinary minimum inhibitory concentration (MIC) testing methods. There have been many studies on the heteroresistance to various antibiotics for specific bacterial species, which also encompass colistin heteroresistance in *K. pneumoniae* isolates [13,14,15,16,17,18]. Although little is known about the clinical significance of antibiotic heteroresistance, instances of treatment failure associated with colistin heteroresistance in *K. pneumoniae* have been reported [19]. In addition, clinicians may be prompted to investigate the most efficient use of colistin on multidrug-resistant *K. pneumoniae*, including antibiotic combinations [20].

In this study, we investigated the incidence rate and genomic variation of colistin heteroresistance in *K. pneumoniae* blood isolates and examined the in vitro efficacy of colistin and meropenem combination treatment against colistin-heteroresistant *K. pneumoniae*.

## 2. Materials and Methods

### 2.1. Bacterial Strains and Antibiotic Susceptibility Testing

In total, 252 nonduplicated *K. pneumoniae* blood isolates were collected from January to December 2017 from Samsung Medical Center (Seoul, Korea). Species identification was performed using a VITEK-2 system (BioMérieux, Hazelwood, MO, USA).

In vitro antimicrobial susceptibility testing was performed using the broth microdilution method outlined in the Clinical and Laboratory Standards Institute (CLSI) guidelines [21]. For all isolates, the MICs of four antibiotic agents—imipenem and meropenem (carbapenems) as well as colistin and polymyxin B (polymyxins)—were determined. The MICs of seven other antibiotics (cefotaxime, ceftazidime, cefepime, ciprofloxacin, amikacin, tigecycline, and piperacillin–tazobactam) were also determined for three colistin-heteroresistant isolates and their resistant subpopulations. Most of the antibiotics, except tigecycline, were purchased from Sigma-Aldrich Corp. (St. Louis, MO, USA), and tigecycline (Tygacil^®^ Injection) was provided from Pfizer (Korea). *E. coli* ATCC 25922 and *Pseudomonas aeruginosa* ATCC27853 were employed as quality control strains.

### 2.2. Detection of Colistin-Heteroresistant Isolates

Colistin heteroresistance was detected using a colistin disk diffusion assay (BD BBL™ Sensi-Disc™ antimicrobial susceptibility test disks, colistin 10 μg) or a colistin E-test strip (bioMérieux SA, France) on Mueller–Hinton agar (Difco BBL, USA) plates, where colonies were observed within the clear zone of inhibition. Subpopulations were separated by subculture, and their colistin MIC was assessed by the broth microdilution method and interpreted according to CLSI guidelines [21]. To confirm the presence of colistin heteroresistance, population analysis profiles (PAPs) were obtained. Full 24 h cultures (~10^8^ CFU/mL) were employed. Bacterial cell suspension samples (50 μL) (corresponding to a 0.5 McFarland standard for *K. pneumoniae* cultures) were plated on Mueller–Hinton agar plates containing 0, 0.5, 1, 2, 4, 6, 8, or 10 mg/L of colistin sulfate (Sigma-Aldrich, St. Louis, MO, USA). After 24 h of incubation at 37 °C, the number of colonies were counted. Colistin heteroresistance was defined as the presence of a colistin-susceptible isolate in which the detectable colistin-resistant subpopulations were able to grow in the presence of ≥10 mg/L of colistin sulfate. The detection limit of colistin-resistant subpopulations was 20 CFU/mL and the lower limit of quantification (LOQ) was 400 CFU/mL (i.e., 2.6 log_10_ CFU/mL) [16].

### 2.3. Genotyping and Sequence Analysis of Genes Associated with Colistin Resistance

Multilocus sequence typing (MLST) was performed for three heteroresistant isolates and their resistant subpopulations using a previously described protocol (www.pasteur.fr/recherche/genopole/PF8/mlstKpneumoniae.html) [22]. Genomic DNAs were isolated from overnight cultures in Luria–Bertani agar at 37 °C using the G-spin™ genomic DNA extraction kit for bacteria G-spin™ Genomic DNA Extraction Mini Kit (for Bacteria)G-spin™ Genomic DNA Extraction Mini Kit (for Bacteria) (iNtRON Biotechnology, Korea).

Polymerase chain reaction (PCR) and DNA sequencing were performed to identify nucleotide and resultant amino acid alterations in PhoPQ, PmrAB, and MgrB of parental colistin-heteroresistant isolates and their resistant subpopulations [23]. The presence of the *mcr*-1 gene was investigated by PCR [24].

### 2.4. Time–Kill Assays

We examined the time–kill kinetics of colistin and/or meropenem against a colistin-heteroresistant *K. pneumoniae* blood isolate (S1703-112), which is also multidrug-resistant. Colistin was added to a logarithmic-phase broth culture of approximately 10^6^ CFU/mL to yield concentrations that were 0-, 0.25-, 1-, and 4-fold of the MIC. Samples were collected at 0, 4, 8, 12, 16, 20, and 24 h after adding antibiotics, and a viable cell count was performed by spirally plating the bacterial cell suspension on Mueller–Hinton agar plates after appropriate dilutions. Time–kill curves were constructed by plotting mean colony counts (log_10_ CFU/mL) versus time. Bactericidal activity was defined as a ≥3 log_10_ CFU/mL reduction in the total CFU/mL from the original inoculum [25]. Synergy was defined as a ≥2 log_10_ CFU/mL decrease between the combination and the most efficient agent alone at 24 h [26].

## 3. Results

Among 252 *K. pneumoniae* blood isolates, 13 and 12 isolates (5.1% and 4.7%) were resistant to colistin and polymyxin B (MICs, >4 mg/L), respectively. Eight and nine isolates (3.2% and 3.6%) showed intermediate resistance toward colistin and polymyxin B. The others (231 isolates, 91.7%) were susceptible to both colistin and polymyxin B. Only three and one isolates were resistant to meropenem and imipenem, respectively, whereas nine and three isolates exhibited intermediate resistance. The others were susceptible to meropenem and imipenem (240 and 248; 95.2% and 98.4%, respectively).

We assayed colistin heteroresistance for 231 susceptible *K. pneumoniae* isolates. As a result, we identified three isolates (1.3%) being heteroresistant to colistin using a disk diffusion test or E-test. They showed typical bactericidal patterns in population analysis profiling (Figure 1A). For each isolate, we obtained two colonies growing within the zone of inhibition in disk diffusion or E-test (Figure 1B). They were separated by subculture and were named RP1 and RP2 after the isolate number. All the colistin-resistant subpopulations of the three heteroresistant isolates showed colistin MICs of ≥64 μg/mL, whereas the colistin MICs of parental *K. pneumoniae* isolates were 0.25 or 1 mg/L (Table 1). The colistin MICs of ≥64 μg/mL in the resistant subpopulations persevered after serial subculture in colistin-free media, indicating their stable feature of colistin resistance. The MICs of the other antibiotics tested in this study were not significantly different, except for cefepime and tigecycline in some resistant subpopulations (Table 1). Particularly, the isolate S1703-112 was nonsusceptible to most antibiotics except gentamicin and tigecycline. Thus, we selected this isolate for time–kill assays to investigate the efficacy of a combination of meropenem and colistin. The isolate S1703-112 produced CTX-M-15, an ESBL. According to MLST analysis, the three colistin-heteroresistant *K. pneumoniae* blood isolates belonged to different clones—ST3217 (S1703-35), ST461 (S1703-109), and ST11 (S1703-112)—which were clones not be strictly associated with colistin resistance. The resistant subpopulations showed the same STs as those of their parental isolates. All isolates were negative for the *mcr-1* gene.

We investigated the amino acid alterations of the two-component regulatory systems PmrAB and PhoPQ, which are known to be associated with colistin resistance in *K. pneumoniae* (Table 2). We identified amino acid variations in 18 sites, where 11 were likely not associated with colistin resistance because the amino acids in the resistant subpopulations could be found in other parental isolates. As a result, it was assumed that seven amino acid substitutions may be associated with colistin resistance in resistant subpopulations: two in PmrA, one in PmrB, two in PhoP, and two in PhoQ. Of note, two resistant subpopulations from the same parental isolate did not show amino acid variations in PmrAB and PhoPQ. Two variations in PhoP (Arg198His and Lys199Asn) and one in PhoQ (Leu414Agr) were identified in S1703-35-RP1 but not in S1703-35-RP2. Further, Asp152Asn in PhoQ was identified only in S1703-35-RP2. For resistant subpopulations of S1703-109, Ile178Phe in PmrA and Asp150Asn in PmrB were found in different resistant subpopulations. In addition, Leu414Agr in PhoQ was identified in S1703-112-RP2 but not in S1703-112-RP1. No changes were found in MgrB.

The time–kill assays were performed for the multidrug-resistant and colistin-heteroresistant *K. pneumoniae* isolate S1703-112. While 4- and 1-fold MICs of meropenem showed complete killing efficacy after 12 and 24 h, respectively (Figure 2A), colistin did not eradicate the colistin-heteroresistant isolate even at 4× MIC (Figure 2B). Although the combination of 0.25× MICs of meropenem and colistin did not kill the heteroresistant isolate, the combination of 1× and 4× MICs demonstrated a rapid killing effect compared with a single regimen of meropenem (Figure 2C).

## 4. Discussion

Heteroresistance has been recognized in both Gram-positive and -negative bacteria and is a phenomenon in which a subpopulation of seemingly isogenic bacteria exhibits a range of susceptibilities to a particular antibiotic [16]. Heteroresistance may have an effect on the outcome of clinical infection, particularly because of limitations in detection by routine microbiological susceptibility testing [12]. This study showed that heteroresistance among apparently susceptible isolates forms a reservoir for the emergence of colistin resistance during treatment.

In this study, only a few *K. pneumoniae* isolates were heteroresistant to colistin. They were clonally unrelated to each other. The rate of colistin heteroresistance found here was lower than that in a previous study [13], in which it was reported that 12 among the 16 colistin-susceptible, carbapenemase-producing *K. pneumoniae* isolates from Greece were heteroresistant to colistin. The rates of colistin heteroresistance vary according to locality, isolation source, treatment of colistin, and so forth. In addition, undetected colistin heteroresistance has been reported in *K. pneumoniae* [19,27], suggesting the possibility that the rate of colistin heteroresistance may be higher than that identified in this study.

We identified amino acid alterations that are supposed to be associated with colistin resistance in resistant subpopulations, but it was not known if the genetic changes were induced by colistin treatment. The amino acid alterations have not been previously reported, and it is not known if the changes affect the function of PmrAB or PhoPQ. Of note, two resistant subpopulations from the same isolate showed different amino acid substitutions in the two-component regulatory systems PmrAB and PhoPQ. To our knowledge, variations between colistin-resistant subpopulations have not been reported thus far. However, diverse genetic variations between colistin-resistant *K. pneumoniae* mutants derived from the same parental strain after treatment have been reported [23]. Our results may indicate that diverse subpopulations with resistance to colistin coexist in the heteroresistant or susceptible isolates, which may develop into resistant strains with diverse mutations associated with colistin resistance.

The combination of meropenem and colistin has been suggested to treat multidrug-resistant *K. pneumoniae* infections [28]. The results of our time–kill assays showed that monotherapy with colistin may be problematic for the treatment of infections caused by colistin-heteroresistant *K. pneumoniae*. Although meropenem alone was effective at killing the heteroresistant isolate, the combination of meropenem and colistin allowed rapid eradication at 1× MICs. Meropenem combined with colistin at the appropriate dosage intervals might be a therapeutic option for infections caused by colistin-heteroresistant *K. pneumoniae*. However, the effectiveness of the combination should be investigated for carbapenemase-producing *K. pneumoniae* isolates.

## 5. Conclusions

We identified three colistin-heteroresistant *K. pneumoniae* isolates. The resistant populations of the same isolate showed different amino acid alterations in PmrAB and PhoPQ. Meropenem combined with colistin would be a suitable therapeutic option for infections caused by multidrug-resistant, colistin-heteroresistant *K. pneumoniae* isolates.

## Figures and Tables

**Figure 1 jcm-08-01444-f001:**
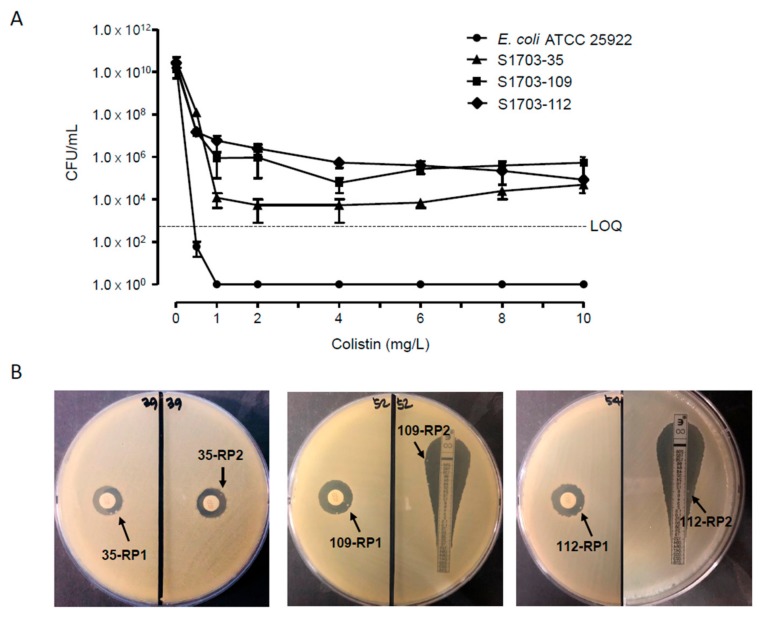
(**A**) Population analysis profiles of three colistin-heteroresistant *Klebsiella pneumoniae* blood isolates and *Escherichia coli* ATCC 25922. LOQ, limit of quantification. The three isolates—S1703-35, S1703-109, and S1703-112—grew in the presence of colistin at concentrations of 4–10 mg/L. (**B**) The results of disk diffusion test or E-test. Resistant subpopulations (each two in three isolates) analyzed further are indicated; 35-RP, 109-RP, and 112-RP indicate the resistant subpopulations of S1703-35, S1703-109, and S1703-112, respectively. For S1703-35, no resistant colonies were detected in the E-test; thus, we selected resistant colonies in independent disk diffusion tests.

**Figure 2 jcm-08-01444-f002:**
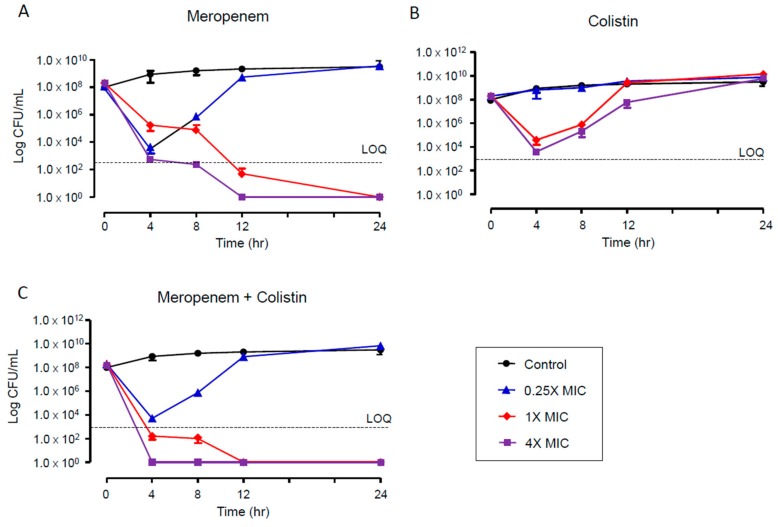
Time–kill curves for meropenem (**A**), colistin (**B**), and combination of meropenem and colistin (**C**) against a colistin-heteroresistant *K. pneumoniae* isolate (S1703-112) that is multidrug-resistant.

**Table 1 jcm-08-01444-t001:** Antibiotic susceptibility against three colistin-heteroresistant *K. pneumoniae* isolates and their resistant populations.

Antibiotics	MIC (mg/L) ^a, b^								
	S1703-35			S1703-109			S1703-112		
	P	RP1	RP2	P	RP1	RP2	P	RP1	RP2
Colistin	1 (S)	128 (R)	64 (R)	0.25 (S)	64 (R)	64 (R)	1 (S)	256 (R)	128 (R)
Polymyxin B	1 (S)	64 (R)	32 (R)	0.25 (S)	64 (R)	32 (R)	1 (S)	64 (R)	64 (R)
Meropenem	0.06 (S)	0.125 (S)	0.125 (S)	0.06 (S)	0.06 (S)	0.06 (S)	4 (R)	4 (R)	2 (I)
Imipenem	1 (S)	1 (S)	0.5 (S)	0.25 (S)	0.25 (S)	0.25 (S)	2 (I)	1 (S)	1 (S)
Cefotaxime	0.125 (S)	0.25 (S)	0.25 (S)	0.25 (S)	0.25 (S)	0.125 (S)	>128 (R)	>128 (R)	>128 (R)
Ceftazidime	0.5 (S)	1 (S)	1 (S)	1 (S)	1 (S)	1 (S)	>64 (R)	>64 (R)	>64 (R)
Cefepime	0.25 (S)	1 (S)	1 (S)	0.125 (S)	1 (S)	1 (S)	>64 (R)	>64 (R)	>64 (R)
Amikacin	4 (S)	4 (S)	4 (S)	2 (S)	2 (S)	2 (S)	32 (I)	32 (I)	32 (I)
Gentamicin	1 (S)	1 (S)	1 (S)	0.5 (S)	0.5 (S)	0.5 (S)	2 (S)	2 (S)	1 (S)
Ciprofloxacin	0.25 (S)	0.25 (S)	0.25 (S)	0.06 (S)	0.06 (S)	0.06 (S)	>64 (R)	>64 (R)	>64 (R)
Aztreonam	0.125 (S)	0.125 (S)	0.125 (S)	0.125 (S)	0.125 (S)	0.125 (S)	>64 (R)	>64 (R)	>64 (R)
Tigecycline	2 (S)	1 (S)	1 (S)	2 (S)	0.5 (S)	0.5 (S)	1 (S)	1 (S)	1 (S)
Piperacillin–tazobactam	16/4 (S)	8/4 (S)	8/4 (S)	8/4 (S)	8/4 (S)	8/4 (S)	>256/4 (R)	>256/4 (R)	>256/4 (R)

^a^ MIC, minimal inhibitory concentration; P, parental; RP, resistant population; S, susceptible; I, intermediate; R, resistant. ^b^ Data are underlined when the MIC increased more than 2-fold in the RP compared with the parental isolate (P).

**Table 2 jcm-08-01444-t002:** Amino acid substitutions in PmrA, PmrB, PhoP, and PhoQ in three colistin-heteroresistant *K. pneumoniae* isolates and their resistant subpopulations.

Isolate^a^		Amino Acid Substitutions in:																	
		PmrA		PmrB		PhoP						PhoQ							
		178	203	43	150	163	185	186	198	199	216	152	154	359	414	421	423	429	430
S1703-35	P	Iso	Arg	Glu	Asp	Arg	Arg	Lys	Arg	Lys	Gln	Asp	Lys	Arg	Leu	Asp	Ala	Val	Phe
	RP1		Lys		Asn		Leu	Glu	His	Asn				Lys	Arg		Pro	Ala	Val
	RP2		Lys		Asn			Glu				Asn		Lys			Pro	Ala	Val
S1703-109	P	Ile	Gly	Arg	Asp	Cys	Thr	Gly	Gly	Cys	Gly	Asp	Ser	Lys	Leu	Gly	Pro	Ala	Val
	RP1	Phe		Glu															
	RP2			Glu	Asn														
S1703-112	P	Ile	Arg			Pro	Leu	Glu	Arg	Lys	Arg	Asp	Gln	Lys	Leu	Gly	Pro	Ala	Val
	RP1		Lys			Arg	Arg				Gln		Lys	Arg					
	RP2		Lys			Arg	Arg				Gln		Lys	Arg	Arg				

^a^ P, parental; RP, resistant population. ^b^ Amino acid alterations that are supposed to be associated with colistin resistance are indicated as white letters with a grey background.

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
