# Peer review of "Colistin Heteroresistance in Klebsiella Pneumoniae Isolates and Diverse Mutations of PmrAB and PhoPQ in Resistant Subpopulations"

_jcm, 2019, doi:10.3390/jcm8091444_

Round 1

Reviewer 1 Report

The paper support the resistance date to colystin but no information are reported to the other antimicrobial molecules.

How  can you explain the low resistance in K. pneumoniae strains in  your geographic area?

Its confirms the genetic changes in K.pneumoniae colistin resistance induced by antimicrobial therapy.

In the materials and methods,  it is not clear if you studied the presence of the ST types in the sensible strains. The ST found belonging to colistin resistant strains can  be  strictly related to resistance or they can easy the resistance acquisition than others ST. 

Considering  the opportunity of combined therapy , it is not described if your indication about the opportunity to use it is associated to your direct experience or to  literature date.  

add to line 39, the reference:

Geraci DM, Bonura C, Giuffrè M, Saporito L, Graziano G, Aleo A, Fasciana T, Di Bernardo F, Stampone T, Palma DM, Mammina C. Is the monoclonal spread of the ST258, KPC-3-producing clone being replaced in southern Italy by the dissemination of multiple clones of carbapenem-nonsusceptible, KPC-3-producing Klebsiella pneumoniae? Clin Microbiol Infect. 2015 Mar;21(3):e15-7. doi: 10.1016/j.cmi.2014.08.022. PMID: 25658574

Bonura C, Giuffrè M, Aleo A, Fasciana T, Di Bernardo F, Stampone T, Giammanco A; MDR-GN Working Group, Palma DM, Mammina C. An Update of the Evolving Epidemic of blaKPC Carrying Klebsiella pneumoniae in Sicily, Italy, 2014: Emergence of Multiple Non-ST258 Clones. PLoS One. 2015 Jul 15;10(7):e0132936. doi: 10.1371/journal.pone.0132936;PMID: 26177547.

and to line 40, the reference :

Mammina C, Bonura C, Di Bernardo F, Aleo A, Fasciana T, Sodano C, Saporito MA, Verde MS, Tetamo R, Palma DM. Ongoing spread of colistin-resistant Klebsiella pneumoniae in different wards of an acute general hospital, Italy, June to December 2011. Euro Surveill. 2012 Aug 16;17(33). pii: 20248; PMID: 22913977.

Author Response

Reviewer 1

The paper support the resistance date to colistin but no information are reported to the other antimicrobial molecules.

- We focused the resistance to colistin. Thus, we did not include the resistance data to other antimicrobial agents except heteroresistant isolates.

How can you explain the low resistance in K. pneumoniae strains in your geographic area?

- In this study, about 5% of K. pneumoniae blood isolated were resistant to colistin. We don’t think that this colistin resistance rate is lower compared with other regions. We think that colistin resistance may have fitness cost, and thus colistin-resistant K. pneumoniae isolates did not disseminate clonally. In addition, low mcr-1 rate would be one of the reasons. 

- In addition, we identified low heteroresistance rate in this study. We assumed that undetected heteroresistance, which is undetected in disc or E-test, would be one of the reasons. We mentioned it in the original manuscript.

Its confirms the genetic changes in K. pneumoniae colistin resistance induced by antimicrobial therapy.

- We identified the genetic changes in resistant subpopulations in heteroresistant K. pneumoniae isolates. However, we did not investigate if the genetic changes were induced by antimicrobial treatment. Of course, genetic changes in PmrAB and PhoPQ after colistin treatment have been reported repeatedly.

“We identified amino acid alterations that are supposed to be associated with colistin resistance in resistant subpopulations, but it was not known that the genetic changes were induced by colistin treatment.”

In the materials and methods, it is not clear if you studied the presence of the ST types in the sensible strains. The ST found belonging to colistin resistant strains can be strictly related to resistance or they can easy the resistance acquisition than others ST.

- We determined the STs in heteroresistant isolates, that is, parental, susceptible isolates in addition to resistant subpopulations. The STs identified in resistant subpopulations were not strictly related to colistin resistance.

“Multi-locus sequence typing (MLST) was performed for three heteroresistant isolates and their resistant subpopulations”

“According to MLST analysis, the three colistin-heteroresistant K. pneumoniae blood isolates belonged to different clones: ST3217 (S1703-35), ST461 (S1703-109), and ST11 (S1703-112), which were clones not be strictly associated with colistin resistance.”

Considering the opportunity of combined therapy, it is not described if your indication about the opportunity to use it is associated to your direct experience or to literature date.

- The combination of meropenem and colistin as therapy to MDR K. pneumoniae infections has been suggested repeatedly. We referred it in the revised manuscript.

“The combination of meropenem and colistin has been suggested to treat multidrug-resistant K. pneumoniae infections [28].”

Crémieux, A.C.; Dinh, A.; Nordmann, P.; Mouton, W.; Tattevin, P.; Ghout, I.; Jayol, A.; Aimer, O.; Gatin, L.; Verdier, M.C.; Saleh-Mghir, A.; Laurent, F. Efficacy of colistin alone and in various combinations for the treatment of experimental osteomyelitis due to carbapenemase-producing Klebsiella pneumoniae. J. Antimicrob. Chemother. 2019, 74, 2666-2675.

Add to line 39, the reference:

Geraci DM, Bonura C, Giuffrè M, Saporito L, Graziano G, Aleo A, Fasciana T, Di Bernardo F, Stampone T, Palma DM, Mammina C. Is the monoclonal spread of the ST258, KPC-3-producing clone being replaced in southern Italy by the dissemination of multiple clones of carbapenem-nonsusceptible, KPC-3-producing Klebsiella pneumoniae? Clin Microbiol Infect. 2015 Mar;21(3):e15-7. doi: 10.1016/j.cmi.2014.08.022. PMID: 25658574.

Bonura C, Giuffrè M, Aleo A, Fasciana T, Di Bernardo F, Stampone T, Giammanco A; MDR-GN Working Group, Palma DM, Mammina C. An Update of the Evolving Epidemic of blaKPC Carrying Klebsiella pneumoniae in Sicily, Italy, 2014: Emergence of Multiple Non-ST258 Clones. PLoS One. 2015 Jul 15;10(7):e0132936. doi: 10.1371/journal.pone.0132936;PMID: 26177547.

and to line 40, the reference :

Mammina C, Bonura C, Di Bernardo F, Aleo A, Fasciana T, Sodano C, Saporito MA, Verde MS, Tetamo R, Palma DM. Ongoing spread of colistin-resistant Klebsiella pneumoniae in different wards of an acute general hospital, Italy, June to December 2011. Euro Surveill. 2012 Aug 16;17(33). pii: 20248; PMID: 22913977.

- As suggested, we included the references in the revised manuscript.

Reviewer 2 Report

The Authors investigated the prevalence of heteroresistance in 252 clinical isolates of Klebsiella pneumoniae against colistin. Among them, three colistin-resistant subpopulations, representing separate clones ST, were selected to further analyses. Genetic analysis revealed the presence different amino acid substitutions in two-component regulatory systems, PmrAB and PhoPQ. The paper is one of several recent ones that describes the phenomenon of heteroresistance that occur in both Gram-positive and Gram-negative bacteria. In general, the manuscript is clear, consistent, with a nicely composed set of interesting data, that in my opinion add to the current state of knowledge. It still contains however some not complete information or statements not justified enough. One of the mechanisms of heteroresistance are spontaneous tandem amplifications, including known resistance genes. In such cases, the heteroresistance trait is usually unstable. Did you determine the stability of colistin-resistant mutants with amino acid changes in the PmrAB or PhoPQ regulatory systems? Secondly, it is known that the amino acids alterations or the substitution positions may affect the function of proteins. Could you discuss what potential consequences may have the amino acids alterations demonstrated in this paper.

Minor comments:

The Material and methods section needs more details. Please provide the companies for the following reagents: antibiotics, bacteriological media e.g. blood agar, Muller-Hinton agar, a colistin E-test strips, and the chemical reagents. Lines 72-74: The procedure „..0.5 McFarland standard for K. pneumoniae cultures grown on blood agar plates for 24 h at 37°C (approximately 108 CFU/ml) were further cultured on Mueller-Hinton agar plates.” is not clear. Could you write it in another way? What method was used to extract bacterial DNA? Please provide the sequences of forward and reverse primers used in this paper (line 90). What is the difference between the detection limit of colistin-resistant subpopulations and the lower limit of quantification (LOQ)? Please describe how it was calculated. Please explain, why and what method was used to determine the mcr-1 gene? (line 92). There is an incompatibility of the method used in this study with the method given in the reference. What criteria were used to categorize the isolates as susceptible, intermediate and as resistant to colistin? (line 106-111) I suggest to add a short description and explain the abbreviations used in Figure 1. The Figure will be clearer. The Y-axis shows probably CFU/ml instead log CFU/ml. Please explain of the reason why the E-test was not used for ST703-35?

Author Response

Reviewer 2

The Authors investigated the prevalence of heteroresistance in 252 clinical isolates of Klebsiella pneumoniae against colistin. Among them, three colistin-resistant subpopulations, representing separate clones ST, were selected to further analyses. Genetic analysis revealed the presence different amino acid substitutions in two-component regulatory systems, PmrAB and PhoPQ. The paper is one of several recent ones that describes the phenomenon of heteroresistance that occur in both Gram-positive and Gram-negative bacteria. In general, the manuscript is clear, consistent, with a nicely composed set of interesting data, that in my opinion add to the current state of knowledge. It still contains however some not complete information or statements not justified enough.

One of the mechanisms of heteroresistance are spontaneous tandem amplifications, including known resistance genes. In such cases, the heteroresistance trait is usually unstable. Did you determine the stability of colistin-resistant mutants with amino acid changes in the PmrAB or PhoPQ regulatory systems?

- We didn’t explore tandem gene amplifications in colistin-heteroresistant isolates. However, we subcultured the resistant subpopulations in colistin-free media, and the resistant feature in the strain were preserved. Thus, we think that the resistance with amino acid changes in PmrAB or PhoPQ would be stable. We mentioned it in the revised manuscript.

“The colistin MICs of ≥64 μg/mL in the resistant subpopulations were persevered after serial subculture in colistin-free media, indicating their stable feature of colistin resistance.”

Secondly, it is known that the amino acids alterations or the substitution positions may affect the function of proteins. Could you discuss what potential consequences may have the amino acids alterations demonstrated in this paper.

- The effect of the amino acid alterations identified in this study was not explored. Thus, we mentioned it in the Discussion section.

“The amino acid alterations have not been previously, and it is not known if the changes affect the function of PmrAB or PhoPQ.”

Minor comments:

The Material and methods section needs more details. Please provide the companies for the following reagents: antibiotics, bacteriological media e.g. blood agar, Muller-Hinton agar, a colistin E-test strips, and the chemical reagents.

- As suggested, we provided the companies.

Most of antibiotics except tigecycline were purchased at Sigma-Aldrich Corp. (St. Louis, MO, USA), and tigecycline (Tygacil® Injection) was at Pfizer (Korea).

colistin E-test strip (bioMérieux SA, France) on Mueller-Hinton agar (Difco BBL, USA) plates

colistin sulfate (Sigma-Aldrich, St. Louis, MO, USA)

Lines 72-74: The procedure „..0.5 McFarland standard for K. pneumoniae cultures grown on blood agar plates for 24 h at 37°C (approximately 108 CFU/ml) were further cultured on Mueller-Hinton agar plates.” is not clear. Could you write it in another way?

- It is not associated with disk or E-test. We moved it to other part.

“Bacterial cell suspension samples (50 μL) (corresponding to a 0.5 McFarland standard for K. pneumoniae cultures) were plated on Mueller-Hinton agar plates”

What method was used to extract bacterial DNA?

“Genomic DNAs were isolated from overnight cultures in Luria-Bertani agar at 37°C using G-spin™ genomic DNA extraction kit for bacteria G-spin ™ Genomic DNA Extraction Mini Kit (for Bacteria)G-spin ™ Genomic DNA Extraction Mini Kit (for Bacteria(iNtRON Biotechnology, Korea).”

Please provide the sequences of forward and reverse primers used in this paper (line 90).

- We used the primer sequences used in our previous study. We referred it.

Kim, S.Y.; Choi, H.J.; Ko, K.S Differential expression of two-component systems, pmrAB and phoPQ, with different growth phases of Klebsiella pneumoniae in the presence or absence of colistin. Curr. Microbiol. 2014, 69, 37-41.

What is the difference between the detection limit of colistin-resistant subpopulations and the lower limit of quantification (LOQ)? Please describe how it was calculated.

- We calculated the LOD (limit of detection) and LOQ (limit of quantification) according to the usual way, that is, LOD is equal to 3 times of standard deviation and LOQ is equal to ten times of standard deviation. In most papers, the method of calculation was not described, and so we did not mentioned it.

Please explain, why and what method was used to determine the mcr-1 gene? (line 92). There is an incompatibility of the method used in this study with the method given in the reference.

- It was our error. We tried to detected mcr-1 by method of PCR. We changed the reference.

Liu, Y.Y.; Wang, Y.; Walsh, T.R.; Yi, L.X.; Zhang, R.; Spencer, J.; Doi, Y.; Tian, G.; Dong, B.; Huang, X.; Yu, L.F.; Gu, D.; Ren, H.; Chen, X.; Lv, L.; He, D.; Zhou, H.; Liang, Z.; Liu, J.H.; Shen, J. Emergence of plasmid-mediated colistin resistance mechanism MCR-1 in animals and human beings in China: a microbiological and molecular biological study. Lancet Infect. Dis. 2016, 16, 161–168.

What criteria were used to categorize the isolates as susceptible, intermediate and as resistant to colistin? (line 106-111)

- The isolates with MICs >4 mg/L were categorized as resistant to colistin.

“Among 252 K. pneumoniae blood isolates, 13 and 12 isolates (5.1% and 4.7%) were resistant to colistin and polymyxin B (MICs, >4 mg/L), respectively.”

I suggest to add a short description and explain the abbreviations used in Figure 1. The Figure will be clearer.

- As suggested, we added a short description and explained the abbreviation (RP).

Figure 1. (A) Population analysis profiles of three colistin-heteroresistant Klebsiella pneumoniae blood isolates and Escherichia coli ATCC 25922. LOQ, limit of quantification. The three isolates, S1703-35, S1703-109, and S1703-112, grew in the presence of colistin at concentrations of 4 to 10 mg/L. (B) The results of disk diffusion test or E-test. Resistant subpopulations, abbreviated as RP (each two in three isolates), analyzed further are indicated.

The Y-axis shows probably CFU/ml instead log CFU/ml.

- We revised it.

Please explain of the reason why the E-test was not used for ST703-35?

- We also tested E-test for S1703-35. However, we did not detect resistant colony in S1703-35 in E-test. Thus, we selected resistant colonies only in disc diffusion method.

“For S1703-35, no resistant colonies were detected in E-test, and thus we selected resistant colonies in independent disk diffusion tests.”